# Determinants of Physical Activity and Screen Time Trajectories in 7th to 9th Grade Adolescents—A Longitudinal Study

**DOI:** 10.3390/ijerph17041401

**Published:** 2020-02-21

**Authors:** Lilian Krist, Stephanie Roll, Nanette Stroebele-Benschop, Nina Rieckmann, Jacqueline Müller-Nordhorn, Christin Bürger, Stefan N. Willich, Falk Müller-Riemenschneider

**Affiliations:** 1Institute for Social Medicine, Epidemiology and Health Economics, Charité-Universitätsmedizin Berlin, Corporate Member of Freie Universität Berlin, Humboldt-Universität zu Berlin, and Berlin Institute of Health, 10117 Berlin, Germany; stephanie.roll@charite.de (S.R.); buergerc@yahoo.de (C.B.); stefan.willich@charite.de (S.N.W.); 2Institute of Nutritional Medicine, University of Hohenheim, 70599 Stuttgart, Germany; N.Stroebele@uni-hohenheim.de; 3Institute of Public Health, Charité-Universitätsmedizin, Corporate Member of Freie Universität Berlin, Humboldt-Universität zu Berlin, and Berlin Institute of Health, 10117 Berlin, Germany; nina.rieckmann@charite.de (N.R.); jacqueline.mueller-nordhorn@charite.de (J.M.-N.); 4Berlin Institute of Health (BIH), Charité-Universitätsmedizin Berlin, 10178 Berlin, Germany; falk.mueller-riemenschneider@nuhs.edu.sg; 5Saw Swee Hock School of Public Health, National University of Singapore; Singapore 117549, Singapore; 6Yong Loo Lin School of Medicine, National University of Singapore, Singapore 117597, Singapore

**Keywords:** physical activity, screen time, trajectory, school students, adolescents, longitudinal

## Abstract

Physical activity (PA) in youth tends to decline with increasing age, while sedentary behaviour including screen time (ST) increases. There are adolescents, however, whose PA and ST do not follow this pattern. The aim of this study is (i) to examine trajectories in PA and ST from grade 7–9 among students in Berlin, and (ii) to investigate the relationship of these trajectories with individual factors and school type. For the present analyses, changes in students’ PA and ST across three time points from 7th to 9th grade were assessed via self-report questionnaires. Positive and negative trajectories were defined for both PA (positive: increasing or consistently high, negative: decreasing or consistently low) and ST (vice versa). Multivariable logistic regression analyses were performed to identify possible predictors of PA and ST trajectories. In total, 2122 students were included (50.2% girls, mean age 12.5 (standard deviation 0.7) years). Compared to grade 7, less students of grade 9 fulfilled PA and ST recommendations (PA: 9.4% vs. 13.2%; ST: 19.4% vs. 25.0%). The positive PA trajectory included 44% of all students (63% boys), while the positive ST trajectory included 21% of all students (30% boys). Being a boy was significantly associated with a positive PA trajectory, while being a girl, having a high socioeconomic status, and attending a high school, were significantly associated with a positive ST trajectory. Different PA and ST trajectories among adolescents should be taken into account when implementing prevention programs for this target group.

## 1. Introduction

The decline in adolescents’ physical activity (PA) and the increase of screen-based sedentary behaviour (screen time; ST) is an important public health concern [1]. While physical activity has a large number of positive effects on physical and psychological health, screen time is associated with an increased risk for cardiovascular and metabolic diseases as well as numerous psychological disorders like depression, low self-esteem, and psychological distress [2,3,4,5,6].

Health behaviours that develop during adolescence can have enduring effects on health behaviours in adult life [7,8]. Movement guidelines for children and adolescents recommend at least 60 min per day of moderate-to-vigorous physical activity (MVPA) and not more than 2 h of recreational screen time per day [9,10,11,12]. The German Children and Youths Survey (KIGGS Wave 1, 2009−2012) reported higher PA at age 11−13 years compared to age 14−17 years (12% compared to 8% of the girls and 17.4% compared to 15% of the boys fulfilled WHO recommendations, respectively) [13]. The German “Health Behaviour in School-aged Children” (HBSC) study (2002−2010) showed higher screen time in adolescents aged 15 years compared to 11 years: 68% compared to 42% of the girls and 67% compared to 50% of the boys, respectively, are spending more than two hours per day on screen time behaviour [14,15].

Evidence from longitudinal studies shows a decline in physical activity and an increase in sedentary behaviour (SB) from childhood through adolescence [8,16,17,18,19,20,21,22]. However, only few longitudinal studies among German adolescents are available. Schwarzfischer et al. reported the longitudinal development of PA and SB at three time points (2008, 2010, and 2013) and associations with BMI and body fat, however, Germany was only one of five European countries included in this study and the measurement of PA was not the study’s main purpose [22]. Braig et al. reported associations of PA and ST with self-esteem in 11−13 year old adolescents in a very selective sample in a small city in southern Germany [3]. Most studies describe socioeconomic status, anthropometric measures, education of parents, and migration background as factors associated with PA and ST [3,23,24]. While high socioeconomic status and high educational levels are mostly associated with high PA and low ST, migration background is more often associated with lower PA and higher ST [25,26]. Regarding school type, no longitudinal studies are available for Germany, but two cross-sectional studies report an association of lower school type with higher screen time behaviour (consumption of TV and computer/gaming consoles) [27,28].

Whereas associated factors are mostly reported for the whole study sample representing average changes of the sample (based on mean differences normally assessed at two time points), this approach does not investigate distinct subgroups following different trajectories of PA and ST [18]. There is growing evidence that health behaviour of children and adolescents is easier understood by taking a deeper look into different subgroups following unique patterns of behaviour change. A recent systematic review of 2019 reported results of 11 studies investigating PA trajectories in a young age group performing descriptive analyses of the emerged trajectories and investigating associations with various predictors and/or determinants (e.g., health behaviour, urban vs. rural dwelling, or parental education) [29]. While trajectory modelling analyses require multiple measurements over a long study period, there are on the other hand studies investigating trajectories with only two assessment points. Two studies (both using predefined trajectories) found associations of built environment, social support, well-being, and parental education with PA trajectories, however, no information about ST trajectories was reported [30,31].

To our knowledge there are no studies investigating the relationship of both socioeconomic and environmental factors with PA and ST trajectories in one study population; and especially in Germany, longitudinal studies investigating PA and ST are lacking. It is still not clearly described to which extent PA decreases and ST increases during adolescence and which percentage of adolescents do not follow this negative trajectory. There is also still a lack of knowledge on the associations of individual and environmental factors with different PA and ST trajectories. Therefore, the aim of this study was to add longitudinal data to the literature by examining patterns of change in PA and ST among school students over two years and to investigate associations of these trajectories with individual and environmental factors (e.g., sex, body mass index (BMI), socioeconomic status, migration background, and school type).

## 2. Materials and Methods

### 2.1. Study Design and Participants

The current analyses are based on two intervention groups of the study sample of the “Berlin Evaluates School Tobacco” (BEST)-prevention study, a three armed cluster randomized controlled trial (RCT) conducted from 2010 to 2014 in Berlin [32,33]. The RCT’s aim was to evaluate a parent-involved smoking prevention for 7th grade students of high schools and integrated secondary schools. Both intervention groups received information about smoking, additionally the parents of one group received information regarding rules to encourage their children not to start or to stop smoking. The intervention did not include any information regarding physical activity or screen time behaviours. The students of the control group, however, received information about nutrition and physical activity and were therefore excluded from the present analyses. Separate signed written informed consent was required from participating students as well as from at least one parent/caregiver. The study was approved by the ethical review committee of the Charité-Universitätsmedizin Berlin, Germany (EA1/133/10).

The present analysis uses cohort type data assessed in the RCT (two intervention groups only). The aim of the present analysis is to investigate health behaviour changes in 7th grade students over two years.

### 2.2. Data Collection

Data were collected at baseline, after 1 and 2 years during classes using self-report questionnaires that were developed on the basis of the standardised and validated questionnaires for adolescents, such as the ones used in the HBSC survey [34,35,36]. Details of the data assessment and a description of the variables were described previously [28,33].

### 2.3. Variable Definitions

#### 2.3.1. Moderate-to-Vigorous Physical Activity

Moderate-to-vigorous physical activity (MVPA) was assessed using three adapted items of the HBSC questionnaire: ‘number of days of physical activity’, ‘weekly hours of physical activity’, and ‘WHO recommendations fulfilled’ (fulfilled if 60 min of physical activity per day were reported).

#### 2.3.2. Screen Time Behaviour

Total recreational screen time comprised watching TV and playing at the computer and/or gaming consoles using items of the HBSC questionnaire. Data for TV and computer use were assessed as ‘hours per weekday’ and ‘hours per weekend day’, respectively. The maximal possible amount of ST that could be reported was 14 h per day; however, we defined 11 h per weekday as maximal plausible amount of ST, while for weekend days we made no changes. According to international guidelines, ‘screen time recommendations fulfilled’ was coded as ‘yes’, if total recreational screen time (TV and computer) was 2 h per day or less [11,12].

#### 2.3.3. Sex, Age, and Anthropometric Data

We assessed sex, age, height, and weight with self-report questionnaires and calculated the body mass index (BMI). According to Cole, we assigned students into four categories, underweight (<10th percentile), normal (10th–<90th percentile), overweight (90th–<95th percentile), and obese (95th percentile or higher) [37,38].

#### 2.3.4. Migration Background

A student was defined as having a migration background if he or she was not born in Germany or if at least one parent was not born in Germany but moved to Germany after 1949 [39].

#### 2.3.5. School Type

There are two school types in Berlin: high schools (Gymnasien) and integrated secondary schools (Integrierte Sekundarschulen). Both schools can be finished with a secondary school leaving certificate after 10th grade or with a high school diploma after 12th grade (high schools) or 13th grade (integrated secondary schools). The academic requirements are higher in high schools than in integrated secondary schools, and the number of school lessons per week differs slightly (33−34 vs. 31−32 lessons per week at high schools and integrated secondary schools, respectively) [40,41]. All schools in Berlin offer three lessons (two hours) of physical education per week according to a central curriculum [42,43].

#### 2.3.6. Socioeconomic Status

The socioeconomic status (SES) was assessed using the family affluence scale (FAS), a validated instrument to assess the material affluence of a family asking for the number of motor vehicles, computers, number of holidays during the last year, and if the child has his/her own bedroom. A score of 0−3 points indicates low affluence, a score of 4−5 middle affluence, and a score of 6−7 high affluence [44].

#### 2.3.7. Parental Work Status

It was assessed via self-report, if both parents work, only one parent, or none of the parents. No information was obtained regarding working hours or workplaces.

#### 2.3.8. Outcomes

The primary outcomes were the PA and ST trajectories (changes) from baseline to 24-month follow-up. We defined four trajectories for PA and ST, respectively. At least 5 h per week of leisure time PA were defined as active taking the 2 h per week of physical education in school into account. For ST, 14 h per week were defined as high screen time behaviour. A relevant change between baseline and 24-month follow-up was defined for PA as a difference of at least 30 min per week and for ST as a difference of at least 3 h per week, reflecting about a tenth of the baseline duration.

We converted the PA trajectories ‘consistently high’ and ‘increasing’ into ‘positive PA trajectory’ and ‘consistently low’ and ‘decreasing’ into ‘negative PA trajectory’. For screen time, ‘consistently low’ and ‘decreasing’ were converted into ‘positive ST trajectory’, while ‘consistently high’ and ‘increasing’ were converted into ‘negative ST trajectory’ (Table 1).

#### 2.3.9. Statistical Analysis

All statistical analyses were performed for the 12–13 years old students due to the small number of students younger than 12 and older than 13 years. We used all data available for the respective analysis; missing data were not imputed. Characteristics of schools and students were analysed by descriptive statistical methods (e.g., mean and standard deviation (SD), frequencies, and percentages) without taking the clustering of classes or schools into account.

For the analyses of associated factors, the four outcome categories (consistently low, decreasing, increasing, and consistently high) were combined into two categories (positive, negative) for PA and ST, respectively. A generalized linear mixed model (GLMM with random intercept and a logit link function) was used taking the nested structure of the data with both fixed and random effects into account. The random factors ‘school’ and ‘class within school’ (as nested factor) were included into the models, while other influencing factors were considered fixed. Results are presented as (adjusted) odds ratios (OR) and 95% confidence intervals (CI). All *p*-values are considered exploratory (no adjustment for multiple testing). Analyses were performed using the software package SAS 9.4 (SAS Institute Inc., Cary, NC, USA) and IBM SPSS Statistics 24 (IBM Corp. Released 2016. IBM SPSS Statistics for Windows, Version 24.0. Armonk, NY, USA: IBM Corp.).

## 3. Results

### 3.1. Participant Characteristics

Out of 2122 students who participated in the baseline assessment, 1795 (84.6%) attended the 24-month follow-up. The baseline sample was 50.2% girls, with a mean age of 12.5 ± 0.7 years. Of all students, 43.3% attended high school, 49.1% had high family affluence, and 34.2% had migration background (Table 2).

### 3.2. PA and ST Changes

From baseline to 24-month follow-up the proportion of students who met the PA recommendations decreased from 13.2% to 9.4% (boys: 16.5% to 12.8%, girls: 10.0% to 6.1%). The proportion of students who reported a PA frequency of at least three times per week decreased, while the proportion of students with a lower frequency increased. In contrast, mean weekly hours of PA were stable over the two-year study period (5.5 ± 5.2 and 5.4 ± 5.4 h) (Table 3).

The proportion of students who fulfilled the screen time recommendations decreased from 25.0% to 19.4% (boys: 17.8% to 12.0%, girls: 31.9% to 26.6%). Total screen time increased by 2.4 ± 19.5 (95% CI: 1.5; 3.4) hours per week (boys: 3.7 ± 19.8 (2.4; 5.1), girls: 1.2 ± 19.2 (0.1; 2.5)). In boys, the increase in screen time was exclusively caused by computer use and gaming, while TV use remained stable. In girls, only TV hours per week showed an increase.

### 3.3. PA and ST Trajectories

Regarding PA, ‘consistently low’ was the largest trajectory, followed by ‘consistently high’, ‘decreasing’, and ‘increasing’. Students in the positive trajectories engaged in about three times as much PA as students in the negative trajectories (Figure 1).

Regarding ST, ‘consistently high’ was by far the largest trajectory, followed by ‘increasing’, ‘consistently low’, and ‘decreasing’. While students in the positive trajectories spent less than 10 h per week in ST behaviour, in the increasing and the consistently high trajectory students spent three to four times the time with ST, respectively. Trajectories were similar for boys and girls (data not shown). Weekly hours spent in PA and ST behaviour are presented in Figure 1.

### 3.4. Factors Associated with PA and ST Trajectories

Regarding PA, the percentage of students in the positive and the negative trajectory was 44% and 56%, respectively. There were more boys than girls (63% vs. 37%) in the positive PA trajectory. School type and socioeconomic factors were similar between the two groups. A slightly higher proportion of students in the positive PA trajectory had higher affluence than middle affluence compared to students in the negative PA trajectory; proportions of low affluence were similar (descriptive results, Table 4).

Regarding ST, only 21% of students followed a positive ST trajectory. There were more girls and high school students in the positive ST trajectory, and family affluence was higher than in the negative ST trajectory (descriptive results, Table 4). All characteristics of the positive and negative PA and ST trajectories are presented in Table 4.

Multivariable regression analyses showed that boys had 2.6 (95% CI: 2.04; 3.30) higher odds to follow a positive PA trajectory than girls; no association was found for SES, school type, migration background, parental work status, BMI, or screen time at baseline (Figure 2).

Girls had a 2.4 (1.82; 3.27) higher odds to follow a positive ST trajectory than boys. Attending a high school (OR 1.59 (1.10; 2.33)) and a high SES (OR 1.99 (1.14; 3.47)) were also associated with following a positive ST trajectory. BMI, migration background, working status of parents, and physical activity at baseline were not associated (Figure 2).

## 4. Discussion

### 4.1. Main Study Findings and Implications

In the overall sample, time spent in moderate-to-vigorous physical activity did not change substantially, while screen time increased over the two-year study period. Regarding the four defined trajectories of PA and ST, respectively, we observed large differences regarding these behaviours. Mean weekly PA was over three times higher among students in the positive PA trajectory than in the negative. In contrast, mean weekly ST was up to four times lower in the positive ST trajectory compared to the negative. These results show that focusing on average values can obscure substantial differences within the target population. Our study shows that boys are more likely to follow a positive physical activity trajectory than girls. A positive screen time trajectory, however, is more often associated with being a girl, having a high socioeconomic status, and attending a high school.

Regarding the longitudinal changes of PA and SB, similar results were reported by Harding et al., who described a substantial increase of SB and only a minimal decrease of MVPA among 12-year-old school students over four years (2008/2009−2012/2013) [45]. One explanation for this observation is that rather light PA (LPA) is replaced by sedentary time, which is often not reported, when the study focused on MVPA, as did our study [46]. Another explanation could be a too short duration of the follow-up. Studies describing a decline of both LPA and MVPA in youth more often have a longer follow-up than the present study and the above-mentioned studies: Dalene et al. reported longitudinal changes of MVPA and sedentary time among a Norwegian youth cohort (age 9−15 years) over six years from 2005/2006 to 2011/2012; however due to only two measurements, no specific age was identified where PA decreased. Ortega et al. stated a linear decline of MVPA among a Swedish and Estonian youth and adolescent cohort (age 8−18 and 15−25 years) which was followed up for 10 years from 1998 to 2008 [8,47]. Despite the vast amount of data, it is still difficult to draw conclusions from the existing studies since results regarding specific age-related PA and ST changes, as well as predictors, vary greatly among different studies [21,48,49].

What our study adds to these longitudinal data is the description of different trajectories for PA and ST. To our knowledge, this is the first German study investigating PA and ST trajectories and their predictors in school students. In addition to the longitudinal design, a novelty of our study is the size of the study sample and the location in a metropolitan area including a high percentage of students with migration background and different levels of socioeconomic status which have been shown to be representative of Berlin school students [28,50].

Regarding PA, only males were associated with a positive PA trajectory. This result is in line with other studies [49,51,52]. For the total sample, no other factors predicted a positive PA trajectory; stratified for sex, however, we found that girls attending a high school were less likely to follow a positive PA trajectory compared to girls attending an integrated secondary school. Similar results were reported in a cross-sectional study (2009/2010) by Czerwinski et al. [25]. This finding may partly be explained by the higher workload of students attending a high school in addition to a higher performance orientation among girls [53].

Following a positive ST trajectory was predicted by several factors. Females were associated with a positive ST trajectory which could be explained by the higher use of electronic gaming among boys, while girls tend to spend their time more often socializing with friends [54,55]. While some studies report a greater increase of SB among girls than boys, that was not the case in our study, probably because we assessed only screen time and not sedentary behaviour [19,56]. High socioeconomic status compared to low SES also predicted a positive ST trajectory. One explanation might be the fact that parents with higher SES have stricter rules regarding gaming and the use of electronic devices [57]. The third factor associated with a positive ST trajectory was attending a high school, probably due to higher parental education among students attending a high school and the somewhat higher proportion of girls among high school students. The number of school lessons differed only slightly between the two school types and does not seem to be a cause for differences in recreational ST time; however, the higher academic requirements among high schools could cause lower screen time in high school students [40,41].

Interestingly, baseline PA was not associated with ST trajectories and baseline ST was not associated with PA trajectories. This is in contrast to a study by Lizandra et al. who illustrated that MVPA was displaced by screen time between 2010 and 2013 [58]. It is possible, however, that ST displaced LPA rather than MVPA which we did not measure individually. In contrast to other studies investigating trajectories, BMI was not associated with PA or ST [30,31].

Health behaviours are complex and even if our study cannot explain causal relationships between the associated variables and the PA and ST trajectories, several implications are drawn from our study results. One implication is to consider that there is no intervention suitable for all boys, all girls, or even all students together. Our study results help to distinguish vulnerable subgroups from the student population and can be transferred to similar contexts as ours, namely metropolitan areas with different types of schools, a broad range of SES, and a high proportion of students with migration background.

To describe the implications more precisely, integrated secondary schools could consider addressing the high screen time behaviour among boys through information campaigns for students including parents as described in a recent systematic review [59], class competitions similar to already existing programs for smoking prevention [60], or extracurricular activities as replacement for gaming at home. Although our study population consisted of high school students, a systematic review showed that school-based PA campaigns should address even younger age groups, include parents, and be monitored carefully [59]. Prevention campaigns for screen time reduction are still scarce. A study by Vik et al. reported no differences in screen time behaviour between the intervention and control group, however, the intervention group showed more positive attitudes towards breaking up sitting time [61]. In addition, since integrated secondary schools are frequented more often by students with a low SES, programs have to consider the lower education and/or fewer resources of the parents, when establishing prevention programs. Kobel et al., for instance, described a positive intervention effect for students having parents with a low educational level [62]. High schools, on the other hand, could focus on physical activity promotion, especially among girls, an approach that Demetriou et al. are planning to use in a study focusing on girls’ PA [63]. A further approach for both types of schools would be to increase the hours of physical education. Our study results add new evidence to the ongoing debate in Germany [64].

### 4.2. Strengths and Limitations

The strengths of the present study include the large sample size, the longitudinal approach of the study, as well as the distribution of migration background, SES, and gender, which appear to be very similar to the student population of Berlin [50]. Another strength is that we took school type into account not only as an associated factor but as a predictor of PA and ST trajectories. Third, we reported PA and ST at the same time and described reciprocal associations; many other studies only focus on one of these two important health behaviours.

Some limitations are considered as well. First, since the aim of the original study was to evaluate smoking behaviour and did not focus primarily on physical activity, PA was not assessed objectively. Self-report of children and adolescents, especially regarding PA, may lead to biased results through misreporting. However, the HBSC questionnaire was validated and used in broad samples of students [35]. Future studies should use accelerometry or other measures to objectively assess PA and ST. Another limitation of the present study is that we assessed screen time only based on the use of TV, computer, and video games as assessed by the HBSC questionnaire [65]. Other increasingly popular screen devices (e.g., smartphones, tablets) were not taken into account, which may have led to an underestimation of ST.

## 5. Conclusions

Among a cohort of 7th grade students that was followed over a two-year period, males were associated with a positive physical activity trajectory (PA remained high or increased), while females, high socioeconomic status, and attending a high school were associated with a positive screen time trajectory (ST remained low or decreased). Taking these results into account could help to tailor prevention programs in order to address special target groups with diverse PA and ST patterns.

## Figures and Tables

**Figure 1 ijerph-17-01401-f001:**
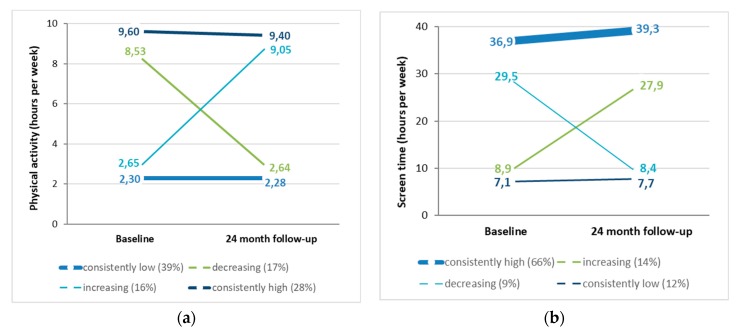
Trajectories (mean hours per week) among 7th grade to 9th grade students for (**a**) physical activity and (**b**) screen time (lines’ weights reflect size of trajectories).

**Figure 2 ijerph-17-01401-f002:**
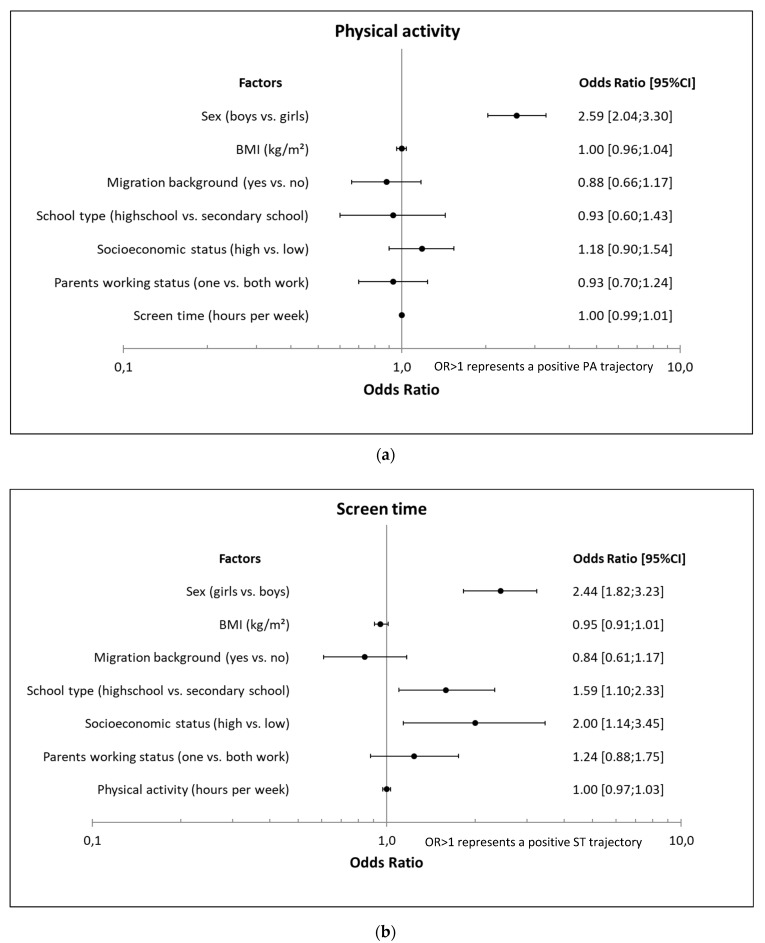
Associations of potential determinants with a positive (**a**) PA (N = 1247) and (**b**) ST (N = 1316) trajectory among 7th to 9th grade students.

**Table 1 ijerph-17-01401-t001:** Definition of physical activity and screen time trajectories.

Trajectory	Baseline	24-Month Follow-Up	Trajectory
**Moderate-to-vigorous physical activity (PA)** **(h per week)**			
Consistently high	≥5	≥5	Positive PA trajectory
Increasing	<5	≥5 ^1^	
Decreasing	≥5	<5 ^1^	Negative PA trajectory
Consistently low	<5	<5	
**Screen time (ST)** **(h per week)**			
Consistently low	≤14	≤14	Positive ST trajectory
Decreasing	>14	≤14 ^2^	
Increasing	≤14	>14 ^2^	Negative ST trajectory
Consistently high	>14	>14	

^1^ Difference between 24-month follow-up and baseline at least 30 min per week; ^2^ Difference between 24-month follow-up and baseline at least 3 h per week.

**Table 2 ijerph-17-01401-t002:** Baseline characteristics of participants.

	Boys	Girls	Total
	N (%) or Mean ± Standard Deviation (SD)		
Students			
Number	1057 (49.8)	1065 (50.2)	2122
Age (years)	12.6 ± 0.7	12.5 ± 0.6	12.5 ± 0.7
Anthropometric data (N = 1895)			
BMI ^1^ (kg/m^2^)	19.0 ± 3.1	18.3 ± 2.8	18.7 ± 3.0
Underweight (BMI < 10th percentile) ^2^	104 (11.0)	186 (19.6)	290 (15.3)
Normal weight (BMI 10th–<90th) percentile) ^2^	686 (72.5)	677 (71.3)	1363 (71.9)
Overweight (BMI 90th–<97th percentile) ^2^	136 (14.4)	79 (8.3)	215 (11.3)
Obesity (BMI ≥ 97th percentile) ^2^	20 (2.1)	7 (0.7)	27 (1.4)
Migration background (N = 1984)			
no	657 (66.2)	649 (65.4)	1306 (65.8)
yes	335 (33.8)	343 (34.6)	678 (34.2)
School type			
High school ^3^ students	417 (39.5)	502 (47.1)	919 (43.3)
Integrated secondary school ^4^ students	640 (60.5)	563 (52.9)	1203 (56.7)
Socioeconomic status (SES)			
Family affluence scale (FAS) (N = 1781)	5.3 ± 1.4	5.1 ± 1.4	5.2 ± 1.4
high (FAS 6–7)	471 (53.2)	404 (45.1)	875 (49.1)
moderate (FAS 4–5)	311 (35.1)	374 (41.7)	685 (38.5)
low (FAS 0–3)	103 (11.6)	118 (13.2)	221 (12.4)
Parent’s working status (N = 1994)			
Both parents work	678 (68.3)	666 (66.5)	1344 (67.4)
One parent works	284 (28.6)	302 (30.1)	586 (29.4)
No parent works	30 (3.1)	34 (3.4)	64 (3.2)

^1^ Body mass index (BMI). ^2^ BMI percentiles according to Cole et al. [37,38]. ^3^ High school: 5th or 7th to 12th grade, highest graduation: high school diploma. ^4^ Integrated secondary school: integration of different school types, 7th to 13th grade, highest graduation: high school diploma.

**Table 3 ijerph-17-01401-t003:** Changes over two years in physical activity (PA) and screen time (ST) behaviour among 7th grade to 9th grade students.

	Boys			Girls			Total		
	N ^1^	Baseline	24-month Follow-Up	N ^1^	Baseline	24-month Follow-Up	N ^1^	Baseline	24-month Follow-Up
Physical activity	(mean ± SD or %)			(mean ± SD or %)			(mean ± SD or %)		
PA at least 60 min/day ^2^	854	16.5	12.8	837	10.0	6.1	1735	13.2	9.4
PA frequency	859			869			1728		
About every day		25.0	21.9		17.8	10.1		21.4	16.0
About 3−5/week		45.8	42.8		36.6	31.5		41.1	37.2
About 1−2/week		23.9	27.0		36.5	42.3		30.2	34.7
About 1−2/month		3.6	5.6		6.4	11.2		5.0	8.4
Never		1.7	2.7		2.6	4.8		2.2	3.8
PA duration Hours per week	742			768			1510		
		6.3 ± 5.9	6.4 ± 5.6		4.6 ± 4.3	4.5 ± 5.0		5.4 ± 5.2	5.4 ± 5.4
Difference between baseline and 24-month follow-up		0.03 ± 7.4			–0.06 ± 5.2			0.015 ± 6.4	
Screen time	(mean ± SD or %)			(mean ± SD or %)			(mean ± SD or %)		
ST ≤ 2 h/day ^3^	853	17.8	12.0	877	31.9	26.6	1730	25.0	19.4
TV (hours/week)	873	17.7 ± 12.3	17.5 ± 12.1	887	15.8 ± 11.7	16.6 ± 11.3	1760	16.8 ± 12.0	17.0 ± 11.7
Computer (hours/week)	862	14.8 ± 12.3	18.8 ± 12.9	887	10.5 ± 11.9	10.9 ± 13.6	1749	12.6 ± 12.3	14.8 ± 13.9
Total screen time Hours/week	853			877			1730		
		31.9 ± 19.5	35.6 ± 18.7		25.9 ± 19.0	27.1 ± 19.1		28.9 ± 19.5	31.3 ± 19.4
Difference between baseline and 24-month follow-up		3.7 ± 19.8			1.2 ± 19.2			2.4 ± 19.5	

^1^ Data of both assessment points available, ^2^ WHO recommendations [10], ^3^ Movement recommendations [9,11].

**Table 4 ijerph-17-01401-t004:** Physical activity and screen time trajectories among 7th grade to 9th grade students.

	PA Trajectories ^1^ (N = 1510)		ST Trajectories ^1^ (N = 1730)	
	Positive	Negative	Positive	Negative
Baseline Variables	%			
Students	44.1	55.9	20.5	79.5
Sex				
Boys	62.9	38.3	29.9	54.3
Girls	37.1	61.7	70.1	45.7
BMI ^2^ categories	(N = 1373)		(N = 1557)	
Underweight (BMI < 10th percentile)^3^	13.5	16.5	21.1	13.5
Normal weight (BMI 10th–<90th percentile) ^3^	75.0	71.6	69.9	73.4
Overweight (BMI 90th–<97th percentile) ^3^	10.4	10.3	8.3	11.5
Obesity (BMI ≥ 97th percentile) ^3^	1.1	1.6	0.6	1.6
Migrant background	(N = 1434)		(N = 1640)	
yes	28.6	33.8	27.6	34.0
School type				
High school students ^4^	46.8	49.9	59.6	42.7
Integrated secondary school students ^5^	53.2	50.1	40.4	57.3
Socioeconomic status (SES)				
Individual SES (family affluence scale; FAS)	(N = 1501)		(N = 1721)	
high (FAS 6−7)	55.4	47.5	61.3	46.2
moderate (FAS 4−5)	33.3	40.3	31.1	40.0
low (FAS 0−3)	11.3	12.2	7.7	13.8
Parents’ working status	(N = 1441)		(N = 1649)	
Both parents work	72.4	69.1	71.5	68.2
One parent works	24.9	27.7	25.3	28.6
No parent works	2.7	3.2	3.2	3.2

^1^ Definition of trajectories is presented in Table 1. ^2^ Body mass index. ^3^ BMI percentiles according to Cole et al. [37,38]. ^4^ High school: 5th or 7th to 12th grade, highest graduation: high school diploma. ^5^ Integrated secondary school: integration of different school types, 7th to 13th grade, highest graduation: high school diploma.

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
