# Peer review of "Determinants of Physical Activity and Screen Time Trajectories in 7th to 9th Grade Adolescents—A Longitudinal Study"

_ijerph, 2020, doi:10.3390/ijerph17041401_

Round 1

Reviewer 1 Report

Thank you for giving me the opportunity to review the manuscript entitled “Determinants of physical activity and screen time trajectories in 7th to 9th adolescents –a longitudinal study”.

The manuscript is addressing an important and actual topic and it is in the scope of the International Journal of Environmental Research and Public Health.

I have included several comments which in my opinion can improve this manuscript.

Introduction

According to the information stated in the introduction, explain the rationale of this study, if it is known that PA decreases with age and sedentary behavior increases, why the population in this study is different from the other studies and also different from the longitudinal studies conducted among German adolescents.

Some data about the prevalence of ST and PA among the study population and in other settings will be useful to have an idea about the magnitude of these conditions in this specific group.

Both PA and ST as the authors have referred are influenced by individual and environmental factors, in this particular case why school type and socioeconomic status would be expecting more related than other factors, evidence summarizing the importance of these or other factors is missing throughout the introduction.

Methods

Include the aim of the cluster randomized controlled trial.

Measurement, include information about validity and reproducibility of the 3 items used to measured MVPA  and screen time behaviors in the study population, and the reference as well

Keep consistency in the terms used in the manuscript, in the introduction authors presented information about sedentary time, while in the methods section the information about data collection is considering screen time behavior and both terms are not synonymous, and it can´t be used interchangeably.

The methods sections and measurement would be improved if variable definitions were organized since the main associations presented in the study aim.

The parental work status variable should include the number of hours and if parents' jobs are conducted outside the home, this approach implies limitations.

Information about the number of hours of a school day and differences in its physical education programs will be necessary to have a higher opportunity to explain or understand the results of this study.

Differences due to a migration status are expecting, however, it wasn´t included information about this association in the introduction section.

Cutoff points of PA and ST to define trajectories describe reasons or the evidence supporting these classifications.

Statistical analysis

% of missing data, in that way multiple imputations should be carried out.

It is important to develop a diagram to explain different sample sizes in the study variables.

Results

It is necessary to specify if data in table 3 is a mean, or %. In addition, include minutes per day or hours per week of PA or sedentary time as characteristics of the study population at baseline.  

Statistical test to express that 44% and 56% is a similar number in PA trajectories.

About Table 3 (PA and ST changes), including a Delta, it means the difference in minutes or hours of PA and ST among waves, to have an idea of the magnitude of the change.

Reviewer 2 Report

Thank you for the opportunity to review your manuscript detailing the physical activity and screen time behaviours of 7th graders over 2 years. 

While I do see the need for such a study in strengthening the evidence, I feel as though your manuscript can go a lot further in adding novel information. Here, I provide the most pressing comments that need addressing:

Abstract: I understand that space is an issue, but I would suggest that you further explain what positive and negative trajectories means.

Introduction: You briefly mention studies by Schwarzfischer and Braig, and I think more detailed explanation of their findings and why they are relevant is necessary. The rationale for this study also needs to be made stronger. By the end of the introduction I am still not sure what this paper is adding, or why you have chosen to specifically focus on SES and school type.

Methods:

No information about ethics has been given

By reducing the trajectories down into two categories a lot of information is being lost. There are much more sophisticated methods of examining trajectories of behaviour, particularly with more than 2 time points. I would strongly encourage you to consider using group based trajectory modelling to examine these trajectories and their predictors.

It is unclear if the screen time measure included just recreational screen time, or all screen time. If it was all screen time, this should be discussed as a limitation as these are school students who would use computers for school related work.

BMI - Why not use conventional cut offs for BMI categories?

What was the rationale for choosing those changes (30 mins for PA and 3 hours for ST) to describe trajectories? 

Analyses - what about data checking and cleaning, and assumptions testing? For ST, the standard deviation is 19.5 hours which seems extremely large. Its necessary to provide information on outliers and how they were assessed in order for readers to know this is a true SD.

Results:

The data presented for tables 2 and 3 is descriptive only. While it is clear that some of the differences between groups are significantly different, it would be useful to provide the actual p-values, or at least p<0.05 or 0.001, etc

Discussion:

The discussion does not clearly articulate the novelty of this research and what it adds to the literature. More thought and time needs to be spent on describing how other researchers/stakeholders can take this information and use it in the future. You mention in the conclusion that the findings can be used to tailor prevention programs, but I think this is already known. More importantly, based on your findings, how can programs be tailored, what strategies are already in place to deliver this, who is best placed to take this forward, and so on.

best of luck and I look forward to another review.
